# NEURAL TEXT ᴅᴇGENERATION WITH UNLIKELIHOOD TRAINING

**Sean Welleck**[1,2*]        **Ilia Kulikov**[1,2*]        **Stephen Roller**[2]        **Emily Dinan**[2]

**Kyunghyun Cho**[1,2,3] **& Jason Weston**[1,2]
[1]New York University, [2]Facebook AI Research, [3]CIFAR Azrieli Global Scholar

## ABSTRACT

Neural text generation is a key tool in natural language applications, but it is well known there are major problems at its core. In particular, standard likelihood training and decoding leads to dull and repetitive outputs (Holtzman et al., 2019). While some post-hoc fixes have been proposed, in particular top-$k$ and nucleus sampling, they do not address the fact that the token-level probabilities predicted by the model are poor. In this paper we show that the likelihood objective itself is at fault, resulting in a model that assigns too much probability to sequences containing repeats and frequent words, unlike those from the human training distribution. We propose a new objective, *unlikelihood training*, which forces unlikely generations to be assigned lower probability by the model. We show that both token and sequence level unlikelihood training give less repetitive, less dull text while maintaining perplexity, giving superior generations using standard greedy or beam search. According to human evaluations, our approach with standard beam search also outperforms the currently popular decoding methods of nucleus sampling or beam blocking, thus providing a strong alternative to existing techniques.

## 1 INTRODUCTION

Neural text generation is a vital tool in a wide range of natural language applications. However, the standard approach – training a sequence to sequence model, e.g. Transformer (Vaswani et al., 2017), to maximize log-likelihood and approximately decoding the most likely sequence – is known to be flawed. Generated text in open-ended applications such as language modeling or dialogue has been observed to be dull, with high frequency tokens used too often and interesting content words used too rarely (Holtzman et al., 2019; Dinan et al., 2019). Moreover, the models repeat themselves at the token, phrase, and sentence levels, and statistics comparing a set of human-generated utterances and model-generated responses indicate a discrepancy between the human and model word distributions. This does not appear to be rectified by training on more data (Radford et al., 2019). Recent fixes involve modifying the decoding strategy using sampling or more sophisticated beam search variants. However, these decoding strategies do not address the core issue: the model's underlying sequence probabilities are clearly not correct.

Several reasons for exactly why neural text is degenerate have been posited, with the cause currently unknown. Possible candidates include the problem being (i) a by-product of the model architecture, e.g. the Transformer architecture preferring repeats (Holtzman et al., 2019; Vig, 2018), (ii) an intrinsic property of human language (Holtzman et al., 2019) rather than a modeling deficiency, or that (iii) a training objective relying on fixed corpora cannot take into account the real goal of using the language (Choi, 2018). Our work shows that, while the above may be factors, a primary factor is the use of the likelihood objective itself, as we demonstrate that degeneration is alleviated if we replace the likelihood objective with our proposal.

While low perplexity in the limit should lead to predicting the correct next target word, there are two major flaws of the likelihood objective: (i) it pays relatively little attention to the argmax or the top of the ranked list of next token probabilities, instead optimizing the likelihood of the entire distribution;

---

*Equal contribution; the ordering was decided by a coin flip.

(ii) it is not focused on optimizing sequence generation, only on producing the next token. The first issue means that greedy or beam search decoding, which rely on the top of the list to generate, are not optimized – there is a discrepancy between maximizing the log-probability of a ground-truth token and ensuring the rank of the ground-truth token to be one. The second issue means that during sequence generation, any imperfection in next token prediction leads to error accumulation that is not addressed by likelihood training.

In this work, we introduce *unlikelihood training*, an approach that addresses the two aforementioned issues. It combines two types of updates: a likelihood update on the true target tokens so that they are assigned high probability, and an *unlikelihood* update on tokens that are otherwise assigned too high a probability. We can collect these *unlikely* token candidates either during next-token prediction or from generated sequences, allowing us to train at both the token and sequence levels. Both token and sequence level unlikelihood training are shown to improve metrics that measure dullness and repetition of the model, while maintaining performance in other metrics such as perplexity or token accuracy compared to the maximum likelihood baseline. Finally, we assess our models using human evaluations. We find that our generations have vastly improved quality compared to likelihood trained models when both models use beam search decoding. Moreover, our approach when using beam search also significantly improves over likelihood trained models using either beam blocking or nucleus sampling, thus outperforming the current state-of-the-art.

## 2 RELATED WORK

**Neural Text Degeneration**   Recently, several papers have observed various forms of *neural text degeneration*, especially in open-ended generation tasks. In dialogue, it has been shown that there is a mismatch between model and human word distributions, where generative models are more likely to output frequent words, but less likely to produce rare words compared to humans. For example, this was observed across all generative models submitted to the ConvAI2 NeurIPS 2018 competition (Dinan et al., 2019). In language modeling, the work of Holtzman et al. (2019) highlighted problems with the word frequency distribution and level of repetition in model generations compared to human text. These issues are not remedied by simply increasing the amount of the training data; e.g. large-scale GPT-2 language models (Radford et al., 2019) display the same issues.

**Improved Decoding Algorithms**   Several methods have been proposed to rectify these issues. The primary ones involve changing the decoding method to a sophisticated beam search variant or to stochastic decoding, e.g. sampling. Different variants of beam search have been explored (Li et al., 2016; Vijayakumar et al., 2018; Kulikov et al., 2018; Holtzman et al., 2018) which can decrease a model's level of repetition by selecting candidates that are unlike previously chosen ones. Separately, hard or soft beam blocking has been investigated (Paulus et al., 2017; Klein et al., 2017), whereby previously generated $n$-grams are blocked from subsequent generation. This approach is often used in dialogue generation, fixing some token or phrase level repetitions but removing repetitions that would naturally occur in human text.

The second major approach is that of sampling from the model at generation time. Top $k$-sampling (Fan et al., 2018) and nucleus sampling (Holtzman et al., 2019) are two methods that sample sequences based on a function of the predicted next token probability distribution given by the model. Both approaches vastly improve the repetition issue, as the randomization often reduces the number of duplicate tokens in a decoded sequence, even if highly scored paths under the model (represented by beam search candidates) contain repetitions. However, as the underlying model is unchanged, it often prefers semantically similar phrasing, depending on the temperature parameter of the sampling (Holtzman et al., 2019). Furthermore, this solution is less relevant in less open-ended tasks such as machine translation, where beam search variants are the preferred method. Ideally we would like a model that can work with both beam and sampling decoding methods.

**Improved Learning Algorithms**   The proposed learning criteria are closely related to structured output prediction methods in which the goal is to increase the scores assigned by a model to true examples while decreasing those assigned to negative examples often generated by the model itself. Some representative algorithms include structured perceptron (Collins, 2002), energy-based models (LeCun et al., 2006) and more recently reflective likelihood (Dieng et al., 2018). A particular variant in this family of algorithms, called negative training, was recently used by He and

Glass (2019) to prevent generic and malicious responses in dialogue models. Similarly, these structured prediction algorithms with neural language models have been applied to machine translation in recent years by Shen et al. (2015) and Edunov et al. (2017).

## 3  NEURAL TEXT GENERATION

**Language Modeling**  In language modeling, our goal is to model a probability distribution $p_*(\mathbf{x})$ over variable-length text sequences $\mathbf{x} = (x_1, \ldots, x_{|\mathbf{x}|})$ composed of tokens from a vocabulary, $x_t \in \mathcal{V}$. We wish to find a model $p_\theta(\mathbf{x})$ which resembles $p_*(\mathbf{x})$, meaning that samples $\hat{x} \sim p_\theta$ are similar to samples from $p_*$, and $p_\theta(\mathbf{x}) \approx p_*(\mathbf{x})$ for all $\mathbf{x}$. When $p_\theta(\mathbf{x})$ is parameterized by a neural network, we call $p_\theta$ a neural language model. We assume that $p_\theta$ takes the form $p_\theta(\mathbf{x}) = \prod_{t=1}^{|\mathbf{x}|} p_\theta(x_t|x_{<t})$.

The *de facto* approach to training such a model is to find parameters $\theta$ that maximize the log-likelihood of a finite set of samples $\mathcal{D}$ from $p_*$ by minimizing:

$$\mathcal{L}_{\mathrm{MLE}}(p_\theta, \mathcal{D}) = -\sum_{i=1}^{|\mathcal{D}|} \sum_{t=1}^{|\mathbf{x}^{(i)}|} \log p_\theta(x_t^{(i)}|x_{<t}^{(i)}). \tag{1}$$

**Sequence Completion**  A closely related problem consists of sampling a sub-sequence, or **prefix**, $\mathbf{x}_{1:k} \sim p_*$, then using $p_\theta$ to conditionally decode a **continuation**, $\hat{\mathbf{x}}_{k+1:N} \sim p_\theta(\cdot|\mathbf{x}_{1:k})$. We now want the resulting **completion** $(x_1, \ldots, x_k, \hat{x}_{k+1}, \ldots, \hat{x}_N)$ to resemble a sample from $p_*$.

We use sequence completion as a setting to study the behavior of neural language models due to its generality. For instance, sequence completion encompasses story generation (Fan et al., 2018), contextual text completion (Radford et al., 2019), language modeling (for $k = 0$), and dialogue modeling (Zhang et al., 2018) where $\mathbf{x}_{1:k}$ is a dialogue history and a continuation is a next utterance.

Given $p_\theta$ and a prefix $\mathbf{x}_{1:k}$, finding the optimal continuation is not tractable, so in practice approximate deterministic or stochastic decoding strategies are used to generate continuations.

**Deterministic Decoding**  Two widely used deterministic decoding approaches are greedy search and beam search. The former can be seen as a special case of the latter.  Greedy search selects the highest probability token at each time step: $x_t = \arg\max p_\theta(x_t|x_{<t})$. Beam search maintains a fixed-size set of partially-decoded sequences, called hypotheses. At each time step, beam search forms new hypotheses by appending each token in the vocabulary to each existing hypothesis, scoring the resulting sequences then selecting the highest scoring sequences. As we describe in Section 4, these deterministic decoding strategies, which depend highly on underlying model probabilities, expose issues with conventionally trained neural language models.

**Stochastic Decoding**  An alternative is to sample from a model-dependent distribution at each step, $x_t \sim q(x_t|x_{<t}, p_\theta)$. In order to prevent sampling low probability tokens, a typical approach is to restrict sampling to a subset of the vocabulary $U \subset \mathcal{V}$ at each step:

$$q(x_t|x_{<t}, p_\theta) = \begin{cases} p_\theta(x_t|x_{<t})/Z & x_t \in U \\ 0 & \text{otherwise,} \end{cases}$$

where $Z = \sum_{x \in U} p_\theta(x|x_{<t})$. The *top-k* sampler restricts sampling to the $k$ most-probable tokens; i.e. $U$ is the size $k$ subset of $\mathcal{V}$ which maximizes $\sum_{x \in U} p_\theta(x|x_{<t})$ (Fan et al., 2018). The *nucleus* sampler instead restricts sampling to the smallest set of tokens with total mass above a threshold $p \in [0, 1]$; i.e. $U$ is the smallest subset with $\sum_{x \in U} p_\theta(x|x_{<t}) >= p$ (Holtzman et al., 2019).

## 4  NEURAL TEXT DEGENERATION

In this section we discuss two degenerate properties that frequently occur in conventional neural language models trained with the maximum likelihood objective (Equation 1).

**Repetition**  First, model-generated continuations exhibit **sequence-level** repetition, especially with deterministic decoding. The problem is seen by observing samples in Appendix Table 4, which

shows completions from the state-of-the-art GPT-2 language model (Radford et al., 2019). Greedy decoding as well as top-k and nucleus sampling exhibit degenerate repetition (with a certain hyperparameter setting), although greedy decoding shows the worst degradation. Using a Transformer language model trained with maximum likelihood (§6), we find that the average percentage of repeated n-grams in model continuations with greedy decoding (43%) far exceeds that of humans (0.5%), computed over prefixes drawn from a validation corpus.

Unlike previous work which only focused on degenerate sequence-level repeats (Holtzman et al., 2019), we additionally observe that neural language models exhibit substantially more repetition in **next-token** prediction compared to human text:

$$\Pr\left(\hat{x}_{k+1} = \arg\max p_\theta(x|\mathbf{x}_{1:k}) \in \mathbf{x}_{1:k}\right) > \Pr\left(x_{k+1} \in \mathbf{x}_{1:k}\right). \tag{2}$$

For instance, the Transformer language model (§6) predicted next-tokens that appeared in the preceding 128 words 62% of the time, versus 49% in ground-truth text. This is especially concerning since the maximum-likelihood objective focuses on optimizing next-token conditional distributions.

**Token Distribution Mismatch**   Second, both greedy continuations and next-token predictions from conventional neural text generators have different token distributions from human text. As demonstrated by Holtzman et al. (2019), such models with greedy or beam search tend to produce high frequency tokens too often and low frequency tokens too rarely, where frequency is defined by the human token distribution. With the Transformer language model (§6), the set of next-token greedy predictions on a held-out validation set had roughly 40% fewer unique tokens than the ground-truth tokens (11.6k vs. 18.9k), and overproduced frequent tokens (Appendix Figure 1). Such behavior has been linked to generations being judged as dull by humans because rare words can add engaging specificity (Weston et al., 2018; See et al., 2019).

## 5   THE UNLIKELIHOOD TRAINING OBJECTIVE

We now describe *unlikelihood training* for neural language models, then in Section 6 demonstrate empirically that our proposal substantially improves neural text degeneration (§4).

### 5.1   UNLIKELIHOOD TRAINING

The key idea behind unlikelihood training is decreasing the model's probability of certain tokens, called *negative candidates*. Given a sequence $(x_1, \ldots, x_T)$ and a set of negative candidate tokens $\mathcal{C}^t = \{c_1, \ldots, c_m\}$, where each $c_j \in \mathcal{V}$, we define the **unlikelihood loss** for step $t$ as:

$$\mathcal{L}_{\text{UL}}^t(p_\theta(\cdot|x_{<t}), \mathcal{C}^t) = -\sum_{c \in \mathcal{C}^t} \log(1 - p_\theta(c|x_{<t})). \tag{3}$$

The loss decreases as $p_\theta(c|x_{<t})$ decreases. We incorporate the unlikelihood loss into a **token-level unlikelihood objective** which augments each time-step of maximum likelihood training:

$$\mathcal{L}_{\text{UL-token}}^t(p_\theta(\cdot|x_{<t}), \mathcal{C}^t) = -\alpha \cdot \underbrace{\sum_{c \in \mathcal{C}^t} \log(1 - p_\theta(c|x_{<t}))}_{\text{unlikelihood}} - \underbrace{\log p_\theta(x_t|x_{<t})}_{\text{likelihood}}. \tag{4}$$

As candidates, we use previous context tokens:

$$\mathcal{C}_{\text{prev-context}}^t = \{x_1, \ldots, x_{t-1}\} \setminus \{x_t\}. \tag{5}$$

Intuitively, minimizing the unlikelihood loss with this candidate set makes (i) incorrect repeating tokens less likely, as the previous context contains potential repeats, and (ii) frequent tokens less likely, as these tokens appear often in the previous context. These candidates are efficient to compute, without requiring additional supervision.

**Gradient analysis**   We assume $p_\theta(x_t|x_{<t}) = \text{softmax}(a)$ and consider the gradient of (4) with respect to the softmax input $a \in \mathbb{R}^{\mathcal{V}}$. With a single negative candidate, the (negative) gradient is:

$$\nabla \mathcal{L}_a = x^* - m \odot p, \quad m_i = \begin{cases} (1 - \alpha \frac{p_{\text{neg}}}{1 - p_{\text{neg}}}) & \text{if } i \neq i_{\text{neg}} \\ (1 + \alpha) & \text{if } i = i_{\text{neg}}, \end{cases} \tag{6}$$

where $x^* \in \{0,1\}^{\mathcal{V}}$ is a one-hot ground-truth vector, $m \in \mathbb{R}^{\mathcal{V}}$, $p = p_\theta(\cdot|x_{<t})$, and $p_{\text{neg}}$ is the probability of the negative candidate at index $i_{\text{neg}}$ (derivation in Appendix A).

This unlikelihood gradient (6) differs from the likelihood gradient, $(x^* - p)$, due to the term $m$ which varies based on the hyper-parameter $\alpha$ and the model's negative candidate probability, $p_{\text{neg}}$. At the ground-truth token index $i^*$, the unlikelihood gradient is positive, increasing the ground-truth token's probability with a magnitude that grows with $p_{\text{neg}}$. Conversely, at the negative candidate index $i_{\text{neg}}$ the gradient is negative. At all other token indices $i \notin \{i^*, i_{\text{neg}}\}$, the gradient moves from negative to positive as $p_{\text{neg}}$ increases. For instance, with $\alpha = 1.0$ the gradient increases the probability of each token $x_i$ when the model assigns high probability to the negative candidate ($p_{\text{neg}} > 0.5$).

## 5.2 Sequence-Level Unlikelihood Training

While the token-level unlikelihood objective efficiently augments maximum likelihood training with token-level penalties, it is limited to prefixes drawn from the training distribution. The resulting *distribution mismatch* between training sequences and generated sequences is a known issue with maximum-likelihood training, motivating objectives that operate on model-generated sequences (Daumé et al., 2009; Ross et al., 2011; Ranzato et al., 2015; Yu et al., 2016).

We thus propose a **sequence-level unlikelihood objective** which uses unlikelihood on decoded continuations. That is, given a prefix $(x_1, \ldots, x_k) \sim p_*$, we decode a continuation $(x_{k+1}, \ldots, x_{k+N}) \sim p_\theta(\cdot|x_1, \ldots, x_k)$, construct per-step negative candidate sets $(\mathcal{C}^{k+1}, \ldots, \mathcal{C}^{k+N})$, and define each per-step sequence-level loss for $t \in \{k+1, \ldots, k+N\}$ as:

$$\mathcal{L}_{\text{ULS}}^t(p_\theta(\cdot|x_{<t}), \mathcal{C}^t) = -\sum_{c \in \mathcal{C}^t} \log(1 - p_\theta(c|x_{<t})). \tag{7}$$

Intuitively, the negative candidates can identify problematic tokens for the loss to penalize. We choose to penalize repeating n-grams in the continuation:

$$\mathcal{C}_{\text{repeat-n}}^t = \{x_t\} \text{ if } (x_{t-i}, \ldots, x_t, \ldots, x_{t+j}) \in x_{<t-i} \text{ for any } (j-i) = n, i \le n \le j, \tag{8}$$

which says that $x_t$ is the (single) negative candidate for step $t$ if it is part of a repeating n-gram[1].

In our experiments we apply this sequence loss in two ways: (i) using it to fine-tune a standard MLE baseline; and (ii) using it to fine-tune an unlikelihood model trained at the token level, $\mathcal{L}_{\text{UL-token}}$. We refer to the former as $\mathcal{L}_{\text{UL-seq}}$ and the latter as $\mathcal{L}_{\text{UL-token+seq}}$. In both cases, fine-tuning is done by equally mixing sequence-level unlikelihood updates (7) and the token-level loss from which it was initially trained (either likelihood updates (1) or token-level unlikelihood updates (4)).

**Efficiency** Any objective that requires explicitly decoding a sequence is constrained by sample efficiency when decoding is slow; if sample efficiency is low, the total decoding time is too large for practical use. In our experiments we show that when used for fine-tuning, the sequence-level unlikelihood objective substantially reduced degeneration in **under 1,500 updates**, rendering it practical for modern large-scale neural models, even with high decoding costs.

## 6 Experiments

We follow a standard language modeling setup from Baevski and Auli (2019) and evaluate our method on the task of sequence completion, detailed below.[2]

**Model Architecture** Recent large-scale language models are based on the Transformer architecture, a multi-layer feed-forward network with self-attention (Vaswani et al., 2017). We use a 16-layer Transformer with 8 attention heads, embedding dimension 1024, and fully-connected dimension 4096; the architecture is based on Baevski and Auli (2019) but with standard embedding and softmax layers. Our proposed method is architecture agnostic; we choose this one as a representative of recent large-scale language models, e.g. Radford et al. (2019).

---

[1]An alternative we tried is to choose a penalization probability $p_{\text{penalize}}$, and use $x_t$ as the single negative candidate for time $t$ when $z_t \sim \text{Bernoulli}(p_{\text{penalize}})$ is 1, and no negative candidate for time $t$ otherwise; this approach was effective but under-performed the $\mathcal{C}_{\text{repeat-n}}$ candidates; see Appendix D.

[2]Code and trained models are available at `https://github.com/facebookresearch/unlikelihood_training`; implemented with Fairseq (Ott et al., 2019).

|  |  | seq-rep-4 |
|---|---|---|
| Prefix | *... Lyrically the song has excerpts of different languages including French , Spanish* | |
| $\mathcal{L}_{\text{MLE}}$ | , Italian , Spanish , Italian , Spanish , Spanish , Portuguese , Portuguese , Portuguese , Portuguese , Portuguese , Portuguese , Portuguese , Portuguese , Portuguese , Portuguese , Spanish , Portuguese , Portuguese , Portuguese , Portuguese , Portuguese | 0.744 |
| $\mathcal{L}_{\text{UL-token+seq}}$ | , and German . In the first verse , the protagonist sings about being a " girl who 's been in love with someone else " , while the second verse describes the relationship between the protagonist and her lover . In the third verse , the protagonist sings | 0.063 |
| Prefix | *... starboard engines and was going to crash . " We 're going in ,"* | |
| $\mathcal{L}_{\text{MLE}}$ | he said . " We 're going to crash . We 're going to crash . We 're going to crash . We 're going to crash . We 're going to crash . We 're going to crash . We 're going to crash . We 're going | 0.787 |
| $\mathcal{L}_{\text{UL-token+seq}}$ | Hood said . " I 'm going to make sure we 're going to get back to the water . " The order to abandon ship was given by Admiral Beatty , who ordered the remaining two battlecruisers to turn away . At 18 : 25 , Hood turned his | 0.000 |
| Prefix | *... career - high 27 points on 8 - for - 11 shooting with three rebounds* | |
| $\mathcal{L}_{\text{MLE}}$ | and two assists . On January 3 , 2012 , he was named to the 2012  13 All - Atlantic 10 first team . On February 3 , 2012 , he was named to the Atlantic 10 first team . On February 5 , 2012 , he was named | 0.277 |
| $\mathcal{L}_{\text{UL-token+seq}}$ | and a career - high 7 assists against the Minnesota Timberwolves . On February 3 , 2012 , he was named to the 2012 All - NBA First Team . On March 7 , 2012 , he was named one of five finalists for the Naismith Award , which is | 0.064 |

Table 1: Example greedy completions showing representative examples of the MLE model's degenerate single-token repetition (top), phrase-level repetition (middle), and 'structural' repetition (bottom), as well as the proposed method's ability to fix these degenerate behaviors.

**Dataset**   We use the Wikitext-103 dataset (Merity et al., 2016), a large-scale collection of Wikipedia articles containing over 100 million words and 260 thousand unique tokens. As a document-level dataset, Wikitext-103 is an open-source representative of recent datasets used for large-scale language modeling (Baevski and Auli, 2019; Radford et al., 2019). We perform experiments at the word level.

**Training**   We train on fixed-length contiguous sequences, in our case of length 1,536, which was selected based on GPU memory constraints. For the token-level losses ($\mathcal{L}_{\text{MLE}}$, $\mathcal{L}_{\text{UL-token}}$), we train each model on 8 GPUs for a maximum of 150k updates, evaluating on the validation set and saving the model state every 10k updates. For the experiments below, we select the saved model state with the best validation perplexity.

Sequence-level fine-tuning begins with the model state selected based on the validation perplexity. Models are fine-tuned for 1,500 total updates. With probability 0.5 an update uses $\mathcal{L}_{\text{ULS}}$ and otherwise uses the token-level loss with which the model was trained. For a $\mathcal{L}_{\text{ULS}}$ update, we split each training sequence and greedily decode continuations (details below). The experiments use a prefix length $k = 50$ and continuation length $N = 100$ for fine-tuning.

**Completions**   We evaluate a model on sequence completion by using the model to decode continuations of prefixes derived from the validation (or test) set. Specifically, the validation (or test) set is first partitioned into sequences of 1,536 tokens, as in training. Then we split each sequence into a batch of prefixes of length $k$ (discarding extra tokens), and decode a continuation of length $N$ for each prefix. The experiments below use $k = 50$ and $N = 100$ for evaluation. For deterministic decoding we use greedy search and beam search with beam size 10, and for stochastic decoding we use top-$k$ sampling with $k \in \{3, 50\}$ and nucleus sampling with $p \in \{0.3, 0.9\}$.

## 6.1   Evaluation Metrics

**Repetition**   As a token-level metric for repetition, we use the fraction of next-token (top-1) predictions that occur in the previous $\ell$ tokens (**rep/$\ell$**); given a set $\mathcal{D}$ of length-$T$ sequences,

$$\text{rep/}\ell = \frac{1}{|\mathcal{D}|T} \sum_{\mathbf{x} \in \mathcal{D}} \sum_{t=1}^{T} \mathbb{I}\left[\arg\max p_\theta(x|\mathbf{x}_{<t}) \in \mathbf{x}_{t-\ell-1:t-1}\right]. \quad (9)$$

A predicted token is called a "single-token repeat" when $\mathbb{I}\left[\cdot\right]$ is 1. Some of these single-token repeats also occur in the human-generated sequences, and we thus report a variant which only counts single-token repeats that are additionally not equal to the ground-truth next-token (**wrep/$\ell$**).

| Model | search | seq-rep-4 | uniq-seq | ppl | acc | rep | wrep | uniq |
|-------|--------|-----------|----------|-----|-----|-----|------|------|
| $\mathcal{L}_{\text{MLE}}$ | greedy | .442 | 10.8k | **25.64** | **.395** | .627 | .352 | 11.8k |
|  | beam | .523 | 9.5k | | | | | |
| $\mathcal{L}_{\text{UL-token}}$ | greedy | **.283** | **13.2k** | 26.91 | .390 | **.577** | **.311** | **12.7k** |
|  | beam | **.336** | **11.7k** | | | | | |
| $\mathcal{L}_{\text{UL-seq}}$ | greedy | .137 | 13.1k | 25.42 | .399 | .609 | .335 | 12.8k |
|  | beam | .019 | 18.3k | | | | | |
| $\mathcal{L}_{\text{UL-token+seq}}$ | greedy | **.058** | **15.4k** | 26.72 | .395 | **.559** | **.293** | **13.8k** |
|  | beam | **.013** | **19.1k** | | | | | |
| Human | - | .006 | 19.8k | - | - | .487 | - | 19.8k |

Table 2: Results for token-level objectives (upper) and sequence-level fine-tuning (lower) according to sequence-level (left) and token-level (right) metrics using the test subset of Wikitext-103.

We use the portion of duplicate $n$-grams (**seq-rep-n**) in a generated sequence to measure sequence-level repetition. That is, for a continuation $\mathbf{x}_{k+1:k+N}$ we compute,

$$\text{seq-rep-n} = 1.0 - \frac{|\text{unique n-grams}(\mathbf{x}_{k+1:k+N})|}{|\text{n-grams}|}, \tag{10}$$

and average over continuations. **seq-rep-n** is zero when the continuation has no repeating n-grams, and increases towards 1.0 as the model repeats. We compute **seq-rep-n** on the continuation.

**Token Distribution**  We quantify a model's predicted token distribution using the number of unique tokens. As a token-level metric (**uniq**), we use the number of unique next-token predictions on a validation or test set $\mathcal{D}$, i.e. $|\{\arg\max p(x_t|x_{<t}) \mid x_{<t} \in \mathcal{D}\}|$. As a sequence-level metric (**uniq-seq**) we use the number of unique tokens in continuations of validation or test prefixes (§6).

**Language Modeling Quality**  We use perplexity (**ppl**), and next-token prediction accuracy (**acc**), defined as $\frac{1}{N}|\{\arg\max p(x_t|x_{<t}) = x_t^* \mid x_{<t} \in \mathcal{D}\}|$, with $N$ prefixes $x_{<t}$ and true next tokens $x_t^*$.

## 6.2  RESULTS

Token-level and sequence-level results on the test set are in Table 2 (valid set in Appendix Table 5).

**Baseline**  The baseline model trained with maximum likelihood ($\mathcal{L}_{\text{MLE}}$) achieved 25.64 test perplexity, comparable to a current state-of-the-art system (Baevski and Auli, 2019) (24.92). However, the greedy baseline's seq-level repeats (seq-rep-4 .442) and single-token repeats (rep .627) far exceed those in human text (.006, .487 respectively). The baseline continuations have far fewer unique tokens than human text (uniq-seq 11.8k vs 19.8k), with a high rate of frequent tokens (Figure 1).

**Token-Level Objective**  The proposed token-level unlikelihood objective ($\mathcal{L}_{\text{UL-token}}$) reduced next-token wrong repetition (wrep .311 vs. .352) while increasing the number of unique next-tokens (uniq 12.7k vs. 11.8k) compared to the baseline ($\mathcal{L}_{\text{MLE}}$). Perplexity and accuracy were similar.

Importantly, the token-level unlikelihood objective yielded substantial improvements in *sequence-level generations*. With greedy search, token-level unlikelihood training improved the 4-gram repetition in continuations by 36% (seq-rep-4 .283 vs. .442) while generating roughly 22% more unique tokens than the baseline (uniq-seq 13.2k vs. 10.8k), and a more favorable rate of infrequent tokens (Figure 1). With beam search, unlikelihood training showed similar improvements over the baseline.

**Sequence-Level Objective**  The sequence level fine-tuning ($\mathcal{L}_{\text{UL-token+seq}}$) yielded further improvements, with a **97% reduction** in 4-gram repetitions (seq-rep-4 .013 vs. .442) from the baseline level (greedy $\mathcal{L}_{\text{MLE}}$), and **77% more** unique tokens (uniq-seq 19.1k vs. 10.8k) with beam search.

Compared to the token-level unlikelihood model ($\mathcal{L}_{\text{UL-token}}$) which was the starting point of fine-tuning, the fine-tuned model's repetition substantially improved (seq-rep-4 .058 vs. .283), unique tokens increased (uniq-seq 15.4k vs. 13.2k), and token-level metrics such as perplexity improved

| Winner | | Loser | Crowdworkers Win rate | Experts Win rate |
|---|---|---|---|---|
| $\mathcal{L}_{\text{UL-token}}$ | | $\mathcal{L}_{\text{MLE}}$ baseline | 57% | |
| $\mathcal{L}_{\text{UL-seq}}$ | | $\mathcal{L}_{\text{MLE}}$ baseline | *71% | |
| $\mathcal{L}_{\text{UL-token+seq}}$ | *beats* | $\mathcal{L}_{\text{MLE}}$ baseline | *82% | |
| $\mathcal{L}_{\text{UL-token+seq}}$ | | $\mathcal{L}_{\text{UL-token}}$ | *75% | |
| $\mathcal{L}_{\text{UL-token+seq}}$ | | $\mathcal{L}_{\text{UL-seq}}$ | 59% | |
| $\mathcal{L}_{\text{UL-token+seq}}$ | *beats* | $\mathcal{L}_{\text{MLE}}$ Nucleus sampling ($p = 0.9$) | 59% | *83% |
| $\mathcal{L}_{\text{UL-token+seq}}$ | | $\mathcal{L}_{\text{MLE}}$ Beam blocking (4-gram) | 60% | *74% |

Table 3: **Human eval results**. * denotes statistical significance (2-sided binomial test, $p < .05$).

(ppl 26.72 vs. 26.91), despite using *only 1,500 updates*. The token distribution improved, with infrequent tokens produced more often than the baseline, and frequent tokens approaching the human level (Figure 1). Finally, after sequence-level fine-tuning, beam search out-performed greedy search.

To visualize how these improvements in metrics translate to generation quality, Table 1 shows greedy completions that characterize the baseline's degeneration and $\mathcal{L}_{\text{UL-token+seq}}$'s improved behavior.

**GPT-2 Fine-Tuning**  In the preceding experiment, sequence-level fine-tuning alone ($\mathcal{L}_{\text{UL-seq}}$) showed substantial improvements over the baseline using a small number of updates. This indicates that the proposed sequence-level fine-tuning can be a cheap, effective way to improve existing pre-trained language models. We demonstrate this by fine-tuning a pre-trained GPT-2 (Radford et al., 2019) language model with sequence-level unlikelihood, using a comparable experimental setup to §6 (details in Appendix C). Fine-tuning with unlikelihood yielded similar improvements in sequence-level repetition (seq-rep-4 .042 vs. .506) to those observed in Table 5, while maintaining language modeling quality according to perplexity and accuracy (see Appendix Table 7).

**Stochastic Decoding**  Although we have focused on deterministic decoding, we also confirm that a model trained with the proposed unlikelihood objectives may still be used with stochastic decoders. Appendix Table 6 shows metrics for completions generated with top-$k$ sampling (Fan et al., 2018) and nucleus sampling (Holtzman et al., 2019). Models trained with unlikelihood objectives maintain language modeling quality compared to the baseline, but with improvements in repetition.

**Human Evaluation**  We perform a crowdworker evaluation to judge the quality of the generations of our proposed models compared to each other, the baseline, two other generation methods, and the reference. We employ a pairwise setup: an evaluator is presented with a prefix and shown continuations from two different models and asked to select which continuation they found more natural. Following Li et al. (2019), we filter workers using quality controls (detailed in Appendix E) and limit the number of annotations that they may complete. Prompts are from the Wikitext-103 test set. All models used beam search (beam size 10) for generation, except for those that use stochastic decoding. We report the win rates for each pairwise comparison.

The main results are presented in Table 3, with additional experiments in Appendix Table 9. We find that all proposed models are preferred over the baseline, and that congruent with automatic metrics, win rates improve after adding the sequence level objective. Our best model also outperforms the baseline used with either nucleus sampling or beam blocking.

We also collected limited annotations from other NLP researchers. These *Expert* annotators were given the same UI as the crowdworkers, and not told about models they were evaluating, but all annotators were familiar with language models. As shown in Table 3, the $\mathcal{L}_{\text{UL-token+seq}}$ model significantly outperforms both nucleus sampling and beam blocking according to the experts.

## 7  CONCLUSION

We described *unlikelihood training*, an approach to training neural language models. We observed that state-of-the-art models trained to maximize likelihood exhibit neural text degeneration, which

we characterized and quantified in terms of repetition and token distribution mismatch. Our results show that the likelihood objective is not constrained enough, in the sense that two models with the same perplexity can exhibit wildly different generation performance. We empirically showed that unlikelihood training - both at the token and sequence levels - substantially reduced degeneration according to automatic metrics, and outperformed likelihood-trained models with various decoding methods according to human evaluation, being superior to the current state-of-the-art approaches.

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

| Prefix | *Lyrically the song has excerpts of different languages including French , Spanish ,* |
|---|---|
| **GPT2-greedy** | German , Italian , Portuguese , Spanish , Portuguese , Portuguese , Portuguese , Portuguese , Portuguese , Portuguese , Portuguese , Portuguese , Portuguese , Portuguese , Portuguese , Portuguese , Portuguese , ... |
| **GPT2-top3** | German , Italian , Portuguese , Spanish , Portuguese , Portuguese, Italian and Spanish. It is also available in English, French and Spanish. In addition to its lyrics, the album features a number of original songs, ... |
| **GPT2-nucleus-0.3** | German , Italian , Portuguese , Spanish , Portuguese , Italian , Portuguese , Spanish , Portuguese , Spanish , Portuguese , Spanish , Portuguese , Spanish , Portuguese , Spanish , Portuguese , Spanish , Portuguese , Spanish , Portuguese , ... |
| **Prefix** | *The first pair of pereiopods is armed with a large , asymm* |
| **GPT2-greedy** | etrical, and long-range laser cannon. The second pair is armed with a large , asymmetrical, and long-range laser cannon. The third pair is armed with a large , asymmetrical, and long-range laser cannon. The fourth pair is armed with a large ... |
| **GPT2-top3** | etrical, and highly mobile head, and the second pair has a small , asymmetrical, and highly mobile head. The second pair has a large and highly mobile head, and the third pair is armed with a large and highly mobile head. The first pair ... |
| **GPT2-nucleus-0.3** | etrical head and a large body. The first pair of pereiopods is armed with a large , asymmetrical head and a large body. The first pair of pereiopods is armed with a large , asymmetrical head and a large body. The first pair of pereiopods is armed ... |

Table 4: Top: Degenerate repetition in completions from a state-of-the-art large-scale language model (Radford et al., 2019). The examples contain single-word repetitions, phrase-level repetitions, and structural repetitions where some tokens within a repeating phrase vary. Recently proposed stochastic samplers (top-$k$, nucleus) exhibit degeneration based on hyper-parameter settings.

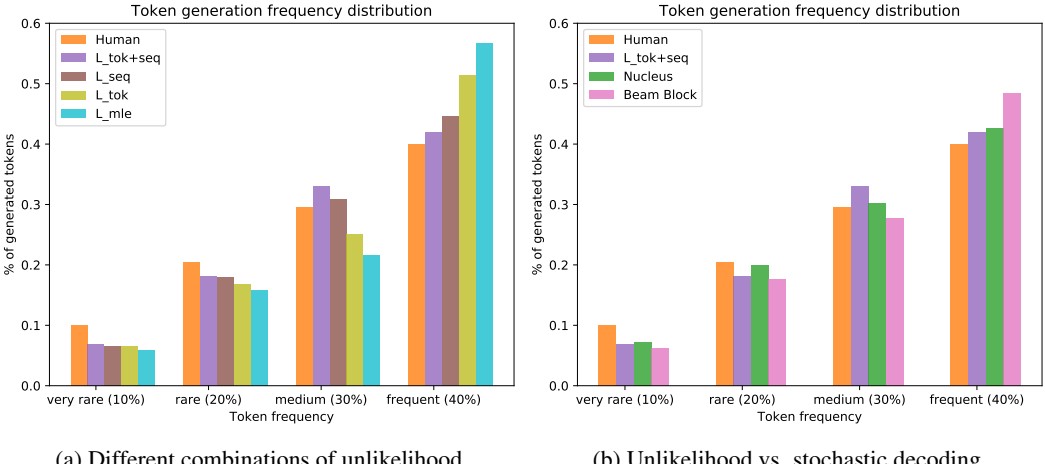

(a) Different combinations of unlikelihood  (b) Unlikelihood vs. stochastic decoding

Figure 1: Sequence-level token distribution using the test subset of Wikitext-103. Nucleus sampling ($p = 0.9$) and beam blocking ($n = 4$) are used with the maximum likelihood baseline ($\mathcal{L}_{\mathrm{MLE}}$).

# A  GRADIENT

**Notation**  Let $x_t^*$ be the true next-token (index $i^* \in \mathcal{V}$) at step $t$, and let $x_{\mathrm{neg}}$ be a negative candidate (index $i_{\mathrm{neg}}$). Let $p = p(x_t | x_{<t}) \in \mathbb{R}^{\mathcal{V}}$ be the output of softmax($a$) where $a \in \mathbb{R}^{\mathcal{V}}$.

Denote the probability of an element $i \in \{1, \dots, V\}$ as $p_i = p(x_t^i | x_{<t})$, and let $p_*$, $p_{\mathrm{neg}}$, and $\tilde{p}_i$ be probabilities of the true next-token, negative-candidate token, and any other token with $\bar{i} \notin \{i^*, \bar{i}\}$.

## A.1  DERIVATION

The (negative) token-level loss with a single candidate is,

$$\mathcal{L}_t = \log p(x_t^* | x_{<t}) + \alpha \cdot \log(1 - p(x_{\mathrm{neg}} | x_{<t})), \tag{11}$$

and its gradient with respect to a logit $a_i$ is:

$$\frac{\partial \mathcal{L}}{\partial p_i}\frac{\partial p_i}{\partial a_i} = (\mathbb{I}[i = i^*] - p_i) - \alpha\frac{p_{\text{neg}}}{1 - p_{\text{neg}}}\left(\mathbb{I}[i = i_{\text{neg}}] - p_i\right). \tag{12}$$

We consider the gradient when $i$ is the true next-token, a negative-candidate, and any other token.

**True Next-Token ($i = i^*$)**

$$\frac{\partial \mathcal{L}}{\partial p_*}\frac{\partial p_*}{\partial a_{i^*}} = (1 - p_*) - \alpha\frac{p_{\text{neg}}}{1 - p_{\text{neg}}}(0 - p_*) \tag{13}$$

$$= 1 - p_*(1 - \alpha\frac{p_{\text{neg}}}{1 - p_{\text{neg}}}). \tag{14}$$

**Negative Candidate ($i = i_{\text{neg}}$)**

$$\frac{\partial \mathcal{L}}{\partial p_{\text{neg}}}\frac{\partial p_{\text{neg}}}{\partial a_{\text{neg}}} = (0 - p_{\text{neg}}) - \alpha\frac{p_{\text{neg}}}{1 - p_{\text{neg}}}(1 - p_{\text{neg}}) \tag{15}$$

$$= -p_{\text{neg}}(1 + \alpha). \tag{16}$$

**Other Token ($i \notin \{i^*, i_{\text{neg}}\}$)**

$$\frac{\partial \mathcal{L}}{\partial \tilde{p}_i}\frac{\partial \tilde{p}_i}{\partial a_i} = (0 - \tilde{p}_i) - \alpha\frac{p_{\text{neg}}}{1 - p_{\text{neg}}}(0 - \tilde{p}_i) \tag{17}$$

$$= -\tilde{p}_i(1 - \alpha\frac{p_{\text{neg}}}{1 - p_{\text{neg}}}). \tag{18}$$

Combining the three cases above, we get:

$$\nabla\mathcal{L}_a = x^* - m \odot p, \tag{19}$$

where $x^* \in \{0, 1\}^V$ is 1 at index $i^*$ and 0 otherwise, and $m \in \mathbb{R}^V$ is:

$$m_i = \begin{cases} (1 - \alpha\frac{p_{\text{neg}}}{1 - p_{\text{neg}}}) & i \neq i_{\text{neg}} \\ (1 + \alpha) & i = i_{\text{neg}}. \quad \square \end{cases} \tag{20}$$

**Multiple Candidates**    In general the objective considers multiple candidates (see section 5):

$$\mathcal{L}_{\text{UL-token}}^t(p_\theta(\cdot|x_{<t}), \mathcal{C}^t) = -\alpha \cdot \underbrace{\sum_{c \in \mathcal{C}^t} \log(1 - p_\theta(c|x_{<t}))}_{\text{unlikelihood}} - \underbrace{\log p_\theta(x_t|x_{<t})}_{\text{likelihood}}. \tag{21}$$

We regroup the token-level objective to be a weighted sum of per-candidate objectives:

$$-\mathcal{L}_{\text{UL-token}}^t(p_\theta(\cdot|x_{<t}), \mathcal{C}^t) = \frac{1}{|\mathcal{C}^t|}\sum_{c \in \mathcal{C}^t}\left(\log p_\theta(x_t|x_{<t}) + \alpha_c \cdot \log(1 - p_\theta(c|x_{<t}))\right) \tag{22}$$

where $\alpha_c = \alpha \cdot |\mathcal{C}^t|$.

Now the gradient can be generalized to multiple candidates, in which case the gradient takes the same form as Eqn. 20, but with $\alpha_c$ in place of $\alpha$.

| Model | search | seq-rep-4 | uniq-seq | ppl | acc | rep | wrep | uniq |
|---|---|---|---|---|---|---|---|---|
| $\mathcal{L}_{\text{MLE}}$ | greedy | .429 | 10.6k | **24.59** | **.401** | .619 | .346 | 11.6k |
|  | beam | .495 | 9.4k |  |  |  |  |  |
| $\mathcal{L}_{\text{UL-token}}$ | greedy | **.274** | **12.6k** | 25.62 | .396 | **.569** | **.305** | **12.5k** |
|  | beam | **.327** | **11.2k** |  |  |  |  |  |
| $\mathcal{L}_{\text{UL-seq}}$ | greedy | .130 | 12.7k | **24.28** | **.406** | .603 | .329 | 12.4k |
|  | beam | .018 | 16.8k |  |  |  |  |  |
| $\mathcal{L}_{\text{UL-token+seq}}$ | greedy | **.051** | **14.8k** | 25.37 | .401 | **.551** | **.287** | **13.4k** |
|  | beam | **.013** | **17.6k** |  |  |  |  |  |
| Human | - | .005 | 18.9k | - | - | .479 | - | 18.9k |

Table 5: Results for token-level objectives (upper) and sequence-level fine-tuning (lower) according to sequence-level (left) and token-level (right) metrics using the validation subset of wikitext-103.

| Search | Model | seq-rep-4 | uniq-seq | ppl | acc | rep | wrep | uniq |
|---|---|---|---|---|---|---|---|---|
| top-k-3 | $\mathcal{L}_{\text{MLE}}$ | .0991 | 14.7k | 25.70 | .350 | .597 | .355 | 12.6k |
|  | $\mathcal{L}_{\text{UL-token}}$ | .0491 | 16.4k | 27.02 | .344 | .539 | .306 | 13.6k |
|  | $\mathcal{L}_{\text{UL-seq}}$ | .0068 | 17.9k | 25.11 | .353 | .581 | .341 | 13.6k |
|  | $\mathcal{L}_{\text{UL-token+seq}}$ | .0087 | 15.2k | 26.84 | .347 | .524 | .292 | 14.6k |
| top-k-50 | $\mathcal{L}_{\text{MLE}}$ | .0165 | 21.9k | 25.70 | .302 | .511 | .303 | 16.1k |
|  | $\mathcal{L}_{\text{UL-token}}$ | .006 | 23.5k | 27.02 | .286 | .440 | .247 | 17.8k |
|  | $\mathcal{L}_{\text{UL-seq}}$ | .0005 | 25.7k | 25.11 | .291 | .497 | .291 | 17.3k |
|  | $\mathcal{L}_{\text{UL-token+seq}}$ | .0009 | 23.7k | 26.84 | .289 | .430 | .238 | 18.8k |
| top-p-0.3 | $\mathcal{L}_{\text{MLE}}$ | .273 | 13.6k | 25.70 | .264 | .339 | .154 | 12.6k |
|  | $\mathcal{L}_{\text{UL-token}}$ | .101 | 16.5k | 27.02 | .247 | .290 | .121 | 13.9k |
|  | $\mathcal{L}_{\text{UL-seq}}$ | .0033 | 20.8k | 25.11 | .266 | .327 | .145 | 13.6k |
|  | $\mathcal{L}_{\text{UL-token+seq}}$ | .0041 | 19.1k | 26.84 | .250 | .284 | .116 | 14.9k |
| top-p-0.9 | $\mathcal{L}_{\text{MLE}}$ | .0154 | 26.9k | 25.70 | .288 | .462 | .263 | 18.6k |
|  | $\mathcal{L}_{\text{UL-token}}$ | .004 | 30.2k | 27.02 | .266 | .381 | .202 | 22.3k |
|  | $\mathcal{L}_{\text{UL-seq}}$ | .0003 | 34.7k | 25.11 | .290 | .450 | .254 | 19.6k |
|  | $\mathcal{L}_{\text{UL-token+seq}}$ | .0007 | 32.4k | 26.84 | .269 | .376 | .198 | 22.7k |
| Human | - | .006 | 19.8k | - | - | .487 | - | 19.8k |

Table 6: Stochastic decoding results according to sequence-level (left) and token-level (right) metrics using the test subset of Wikitext-103.

# B  Stochastic Decoding Results

Table 6 provides automatic metrics for top-$k$ and nucleus sampling (called top-$p$) on the Wikitext-103 test set. These can be compared with the main results of the paper in Table 2. In general, sampling methods yield worse next-token predictions than deterministic approaches (0.302 vs. 0.394 acc for top-k-50 vs. greedy MLE, where acc for stochastic decoding measures the probability that the decoding strategy chooses the ground truth word given a ground truth context). As the choice of sampling hyperparameter gets closer to greedy (i.e. lower values of $k$ and $p$) next token accuracy improves, eventually approaching the greedy MLE results. The unlikelihood-trained sampling models have similar next token accuracy (acc) to their likelihood-trained counterparts, but exhibit fewer repetitions. For lower values of $p$ and $k$ the improvements of unlikelihood training are larger, e.g. 0.277 reduced to 0.0041 for 4-gram sequence repetitions (seq-rep-4) using top-p-0.3. At higher levels of $p$ and $k$, for all methods the continuations contain more unique tokens than that of humans, meaning those values may be too high.

| Model | search | seq-rep-4 | ppl | acc | rep | wrep | uniq |
|---|---|---|---|---|---|---|---|
| GPT-2 | greedy | .506 | 20.75 | .430 | **.589** | .306 | **13.3k** |
| GPT-2$_{\text{MLE}}$ | greedy | .460 | **15.82** | **.464** | .612 | **.305** | 11.8k |
| GPT-2$_{\text{UL-seq}}$ | greedy | **.042** | 18.49 | .444 | .613 | .317 | 11.3k |
| Human | - | .005 | - | - | .407 | - | 17.7k |

Table 7: GPT-2 results according to sequence-level and token-level metrics using the validation subset of wikitext-103. seq-rep-4 is computed on the word level; ppl, acc, rep, wrep are computed on the BPE level.

## C  GPT-2 FINE-TUNING

We evaluated the GPT-2 medium pre-trained model ('GPT-2') and two separate fine-tuning variants on Wikitext-103. The first variant ('GPT-2$_{\text{MLE}}$') was fine-tuned using maximum likelihood; we select the model state with the lowest validation perplexity. The second model ('GPT-2$_{\text{UL-seq}}$') was fine-tuned using the sequence-level unlikelihood objective (§5.2). For both evaluation and sequence-level tuning, we used a prefix length of 50 BPE tokens and a continuation length of 100 BPE tokens. In order to train on a single GPU, we used a batch-size of 1024 tokens for MLE updates, and 300 prefix tokens for unlikelihood updates. Due to the smaller batch size and single-GPU setting, we used 10,000 updates during sequence-level fine-tuning, comparable to the 1,500 updates in the main experiment (§6) in terms of the total number of tokens. Results are shown in Table 7.

## D  SEQUENCE-LEVEL RANDOM CANDIDATES

In Sec. 5.2 we described a way to penalize tokens that occurred in a n-gram repetition. One alternative is to penalize a random subset of the generated sequence. That is, given a continuation $x_{t+1}, \ldots, x_{t+K}$, we now define per-step candidates $(\mathcal{C}^{k+1}, \ldots, \mathcal{C}^{k+N})$ as:

$$\mathcal{C}^t_{\text{random-seq}} = \begin{cases} \{x_t\} & \text{if } z_t = 1 \\ \emptyset & \text{if } z_t = 0, \end{cases} \tag{23}$$

for each $t \in \{k+1, \ldots, k+N\}$, where $z_t \sim \text{Bernoulli}(p_{\text{penalize}})$, and $p_{\text{penalize}} \in [0, 1]$ is a fixed hyper-parameter. Intuitively, these candidates identify random tokens in the generated sequence (hence 'random-seq'), which are then penalized by the sequence-level loss (Eqn. 7).

Results with different values of $p_{\text{penalize}}$ are shown in Table 8. Penalizing 10% of the generated tokens led to substantial improvements in seq-rep-4 for both greedy and beam search compared to the baseline (e.g. 41% for $\mathcal{L}_{\text{UL-seq}}$ greedy, 73% for $\mathcal{L}_{\text{UL-tok+seq}}$ greedy), though using n-gram repetition candidates yielded further improvements (§5.2, Table 5). Improvements in single-token metrics were similar to those from the n-gram repetition candidates (e.g. wrep .287). These results with random-seq candidates demonstrate that sequence fine-tuning can yield improvements without explicitly using the notion of repetition for candidate selection. We also find that penalizing 90% of the generated tokens yields substantial improvements in beam search, but not greedy search; investigating this is left as future work.

| Model | $p_{\text{penalize}}$ | search | seq-rep-4 | uniq-seq | ppl | acc | rep | wrep | uniq |
|---|---|---|---|---|---|---|---|---|---|
| $\mathcal{L}_{\text{MLE}}$ | - | greedy | .429 | 10.6k | 24.590 | .401 | .619 | .346 | 11.6k |
|  | - | beam | .495 | 9.4k |  |  |  |  |  |
| $\mathcal{L}_{\text{UL-seq}}$ | 0.1 | greedy | .253 | 9.9k | 24.329 | .404 | .602 | .330 | 12.3k |
|  |  | beam | .274 | 13.1k |  |  |  |  |  |
| $\mathcal{L}_{\text{UL-seq}}$ | 0.9 | greedy | .434 | 5.3k | 26.519 | .399 | .600 | .330 | 12.2k |
|  |  | beam | .231 | 13.5k |  |  |  |  |  |
| $\mathcal{L}_{\text{UL-tok+seq}}$ | 0.1 | greedy | .116 | 12.5k | 25.518 | .399 | .551 | .287 | 13.2k |
|  |  | beam | .146 | 14.2k |  |  |  |  |  |
| $\mathcal{L}_{\text{UL-tok+seq}}$ | 0.9 | greedy | .423 | 6.7k | 26.629 | .396 | .551 | .288 | 13.2k |
|  |  | beam | .080 | 16k |  |  |  |  |  |
| Human | - | - | .005 | 18.9k | - | - | .479 | - | 18.9k |

Table 8: Results for sequence-level fine-tuning using **random-seq candidates** according to sequence-level (left) and token-level (right) metrics using the **validation subset of wikitext-103**.

# E  HUMAN EVALUATION DETAILS

## E.1  UI SCREENSHOT

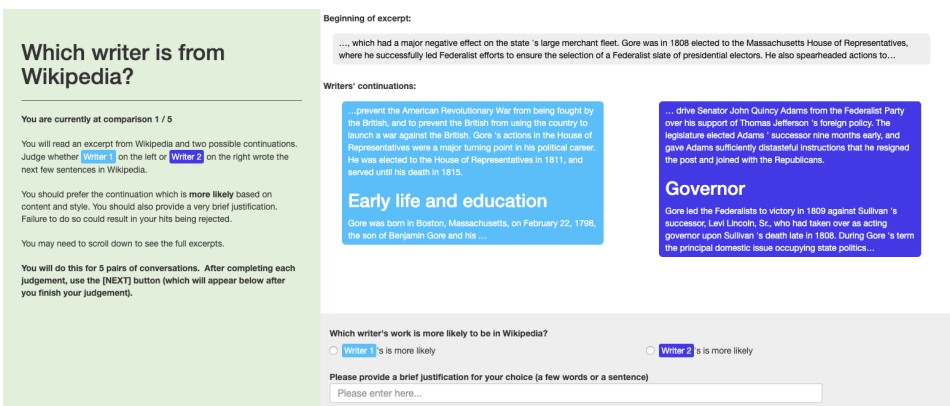

Figure 2: Screen shot of the user interface used in the human evaluation.

## E.2  CROWDWORKER QUALITY CONTROLS

We require workers to correctly answer both of the following quality control questions for their evaluations to be included. Both quality controls compare the true completion against a greedy baseline model.

Following Li et al. (2019), we informed workers that they must provide reasoning for their choices. We filtered workers who did not provide reasoning for at least 80% of their choices.

63% of workers fail at least one of our three quality control mechanisms (2 quality control metrics, and failing to give reasons). 61% fail at least one quality control question; 16% of workers fail both; 4% of workers fail to give reasoning for their choices.

### E.2.1  QUALITY CONTROL 1

**Prompt**  = = In the decades since its release , The Hustler has cemented its reputation as a classic . Roger Ebert , echoing earlier praise for the performances , direction , and cinematography and adding laurels for editor Dede Allen , cites the film as " one of"'

**Correct answer**  those films where scenes have such psychic weight that they grow in our memories . " He further cites Fast Eddie Felson as one of " only a handful of movie characters so real that the audience refers to them as touchstones . " TV Guide calls the film a " dark stunner " offering " a grim world whose only bright spot is the top of the pool table , yet [ with ] characters [ who ] maintain a shabby nobility and grace . " The four leads are again lavishly praised for their performances and the

**Incorrect answer**  the most influential films of the year " . In his review for the Chicago Sun @-@ Times , Richard Corliss calls it " a film of the highest order " and " a film of the highest order " . In his review for the Chicago Sun @-@ Times , Richard Corliss calls it " a film of the highest order " and " a film of the highest order " . In his review for the Chicago Sun @-@ Times , Richard Corliss calls it " a film of the highest order " and " a film of

### E.2.2  QUALITY CONTROL 2

**Prompt**  , which had a major negative effect on the state 's large merchant fleet . Gore was in 1808 elected to the Massachusetts House of Representatives , where he successfully led Federalist efforts to ensure the selection of a Federalist slate of presidential electors . He also spearheaded actions to

**Correct answer**  drive Senator John Quincy Adams from the Federalist Party over his support of Thomas Jefferson 's foreign policy . The legislature elected Adams ' successor nine months early , and gave Adams sufficiently distasteful instructions that he resigned the post and joined with the Republicans . = = Governor = = Gore led the Federalists to victory in 1809 against Sullivan 's successor , Levi Lincoln , Sr. , who had taken over as acting governor upon Sullivan 's death late in 1808 . During Gore 's term the principal domestic issue occupying state politics

**Incorrect Answer**  prevent the American Revolutionary War from being fought by the British , and to prevent the British from using the country to launch a war against the British . Gore 's actions in the House of Representatives were a major turning point in his political career . He was elected to the House of Representatives in 1811 , and served until his death in 1815 . = = Early life and education = = ¡/s¿ ¡/s¿ Gore was born in Boston , Massachusetts , on February 22 , 1798 , the son of Benjamin Gore and his

### E.3  FULL HUMAN EVALUATION RESULTS

| | | Crowdworkers | | Experts | |
|---|---|---|---|---|---|
| Winner | Loser | Win rate | W–L | Win rate | W–L |
| $\mathcal{L}_{\text{UL-token}}$ | $\mathcal{L}_{\text{MLE}}$ baseline | 57% | 17–13 | | |
| $\mathcal{L}_{\text{UL-seq}}$ | $\mathcal{L}_{\text{MLE}}$ baseline | *71% | 41–17 | | |
| $\mathcal{L}_{\text{UL-token+seq}}$ *beats* | $\mathcal{L}_{\text{MLE}}$ baseline | *82% | 41–9 | | |
| $\mathcal{L}_{\text{UL-token+seq}}$ | $\mathcal{L}_{\text{UL-token}}$ | *75% | 56–19 | | |
| $\mathcal{L}_{\text{UL-token+seq}}$ | $\mathcal{L}_{\text{UL-seq}}$ | 59% | 38–27 | | |
| $\mathcal{L}_{\text{UL-token+seq}}$ *beats* | Nucleus | 59% | 47–33 | *83% | 30–6 |
| $\mathcal{L}_{\text{UL-token+seq}}$ | Beam blocking | 60% | 50–34 | *74% | 25–9 |
| Reference | $\mathcal{L}_{\text{MLE}}$ baseline | *85% | 17–3 | | |
| Reference | Nucleus | *69% | 38–17 | | |
| Reference | Beam blocking | *68% | 48–23 | | |
| Reference *beats* | $\mathcal{L}_{\text{UL-token}}$ | *73% | 44–16 | | |
| Reference | $\mathcal{L}_{\text{UL-seq}}$ | 50% | 30–30 | | |
| Reference | $\mathcal{L}_{\text{UL-token+seq}}$ | *64% | 46–26 | | |

Table 9: **Full human evaluation results**. Includes additional comparisons omitted for brevity, and the raw number of wins and loses by each comparison.

