# OpenReview forum: "Neural Text Generation With Unlikelihood Training"
_ICLR.cc/2020/Conference — Accept (Poster)_

### Official Review · AnonReviewer3 · 2019-10-22
**Official Blind Review #3**

**Rating:** 6

**Review:**


This paper proposes training losses, unlikelihood objective, for mitigating the repetition problem of the text generated by recent neural language models. The problem is well-motivated by evidence from the existing literature. Specifically, the paper argues that the main cause of the degenerated output is the maximum likelihood objective commonly used to train language models. Their main contribution is to introduce additional objectives to penalize “unlikely” word probabilities. The proposed penalty is derived into 2 objectives: token level (previous words in context) and sentence level (future decoded words). The prior objective is used along with the MLE, while the later and more expensive is used for fine-tuning. They perform experiments on Wikitext-103 and evaluate models on the perplexity of the models, and n-gram statistics such as repetition, and uniqueness of the decoded texts. The proposed training scheme (UL-token+seq) is shown to have the closest statistics to the original corpus while the perplexity slightly suffers. The additional manual analysis shows that human annotators prefer the outputs (sentence completion) of the proposed method over the other baselines.

Overall, this paper tackles a relevant problem and could propose a novel method.

For the unlikelihood objectives, there are a few clarifications on the design decision. There are some “correct” repetitions in the ground-truth text as well. However, the proposed loss minimizes all repetitions regardless. In addition, it is unclear how the proposed method mitigates the token distribution mismatch. Finally, there is a similar work where they attempt to “match” repetition (or n-gram distribution) of a reference corpus (https://www.aaai.org/ocs/index.php/AAAI/AAAI18/paper/view/16961). A discussion of how this paper distinguishes from previous work would be helpful.

For the experiments, there are some missing detail and concerns:

1. The fine-tuning using UL-seq (eq 7) procedure is not well explained. For example, how many sequences are used per update? How many times that you decode in the 1,500 updates?

2. the stochastic decoding results are related as the paper motivation is built on top of Holtzman et al., 2019, the results also support the claim. However, how do you explain the discrepancy between the experts and the crowd workers?

3. GPT-2 results show that UL-seq does not significantly improve several evaluation metrics from MLE. This is a conflict with the result in Table 2. This could be a sign of the generalization problem of the proposed method. Additional results on different model architectures would be helpful.

Minor question:
1. It is uncertain how these repetition and uniqueness statistics translate to a downstream task (e.g. summarization or NMT). Do you have results regarding this?

2. It appears that UL-token does not do much. What is the increase in training time? Do you recommend using the token loss?


**Experience Assessment:**

I have published one or two papers in this area.

**Review Assessment: Checking Correctness Of Derivations And Theory:**

I assessed the sensibility of the derivations and theory.

**Review Assessment: Checking Correctness Of Experiments:**

I carefully checked the experiments.

**Review Assessment: Thoroughness In Paper Reading:**

I read the paper at least twice and used my best judgement in assessing the paper.

---

> ### Author Response · Authors · 2019-11-12
> **Response to Reviewer #3**
>
> We thank the reviewer for the review; the primary comments were clarifications and the degree of improvement with GPT-2, which we address below.
>
> > “There are some “correct” repetitions in the ground-truth text as well. However, the proposed loss minimizes all repetitions regardless.”
>
> The token-level loss does not penalize ground-truth repetitions, as the ground-truth next-token is not a candidate (due to the \ {x_t} in equation 5).
>
> For the sequence-level loss, the sequence is model-generated, so there is no notion of ground-truth. We also showed that penalizing randomly selected tokens gave improvements (Table 8), so the penalties do not need to be tied to the notion of repetition.
>
> We track the wrep metric which only counts _incorrect_ repetitions (as opposed to rep which counts all repetitions). You can see in Table 2 that wrep improves after unlikelihood training.
>
> > “It is unclear how the proposed method mitigates the token distribution mismatch.”
>
> If frequent words are appearing too often in the decoded sequences then they are the words being penalized with the unlikelihood loss, which naturally shifts mass to less frequent words. Reducing repetitions towards the human rate also results in a closer distribution match. Together, it seems these effects do indeed mitigate token distribution mismatch, as shown in Figure 1.
>
> > “there is a similar work where they attempt to “match” repetition (or n-gram distribution) of a reference corpus (https://www.aaai.org/ocs/index.php/AAAI/AAAI18/paper/view/16961).”
>
> We will add this citation to the related work. The method there maintains a pool of generated text to estimate model marginals, which is a different approach than unlikelihood. Moreover, they only consider local repetition within a 3-word window, which does not address structural repetition or phrase-level repetition for a sufficiently long phrase (e.g. see Table 1).
>
> > “How do you explain the discrepancy between the experts and the crowd workers?”
>
> Crowdworkers and experts all agree that our methods outperform existing approaches, while experts are more confident. This makes sense because crowdworkers are less experienced in this task, and more likely to tend closer to unsure (closer to 50%).
>
> > “GPT-2 results show that UL-seq does not significantly improve several evaluation metrics from MLE.”
>
> The token-level ppl and acc metrics improve over zero-shot GPT-2, while substantially decreasing sequence repetition (over both zero-shot GPT-2 and fine-tuned GPT-2_MLE).
>
> Our primary experiments are conducted with the (Baevski and Auli) model since it allowed us to train from scratch. The GPT-2 results show that unlikelihood is also applicable in a setting where the model is trained on a corpus different from the one it is fine-tuned on; further investigating these variants is interesting future work.
>
> > “It is uncertain how these repetition and uniqueness statistics translate to a downstream task”
>
> Indeed, this is interesting as future work to apply to further downstream tasks. We focus on proposing the general framework here and thoroughly investigating the completion task, which had been previously shown in Holtzman et al 2019 to be an open problem that should be tackled.
>
> > “It appears that UL-token does not do much. What is the increase in training time? Do you recommend using the token loss?”
>
> Yes, we recommend using the token loss. Considering that UL-token uses the same amount of supervision as normal MLE training (i.e. previous-context candidates are already present in the training sequence), the ~16-20% improvement in repetition (Table 2) is substantial. We also see that token+seq is better than only seq (Table 2). There is only a slight increase in training time and it is not a significant limiting factor since the candidates can be computed efficiently.

---

### Official Review · AnonReviewer2 · 2019-10-22
**Official Blind Review #2**

**Rating:** 6

**Review:**


Contributions:

The main contribution of this paper lies in the proposed unlikelihood training objective for open-ended text generation. The key idea is to enforce the unlikely generations to be assigned lower probability by the model. Both token and sequence-level unlikelihood training objectives are provided. Impressively, the authors show that models trained with the proposed method can generate high-quality text via only beam search, without using top-k, nucleus sampling, or beam blocking methods.

Strengths:

(1) Writing & Clarity: The proposed model is very well motivated, the paper is well written, and clearly presented. I enjoyed reading the paper.

(2) Novelty: Though the proposed model is simple, I think it has novelty inside. The proposed model makes connection to negative sampling, and is very intuitive to reduce repetition during the training stage, instead of decoding stage. I find the gradient analysis in Section 5.1 is especially interesting.

(3) Experiments: The authors did a careful job in experiments design, and conducting the experiments. Human evaluation is also provided. A lot of additional results are provided in Appendix. I feel the experiments are solid and convincing.

Weaknesses:

(1) Clarity: I have three questions regarding this paper.

a) Using previous generated tokens as the unlikely tokens for the current generation step in the token-level unlikelihood training is intuitive, but also seems too simple. Can the authors provide some comments on this? Or, are there any better designs?

b) How is the training looking like? Do we need Gumbel-softmax-like trick to backpropagate through the generated tokens in the sequence-level training? Or, this is not needed? Can the authors clarity the training process?

c) The proposed model can be directly applied to dialog response generation task, which also requires diversity in generated responses. Any reason why this conditional generation task is not performed? Or, do the authors plan to also apply the proposed method to this application?







**Experience Assessment:**

I have published one or two papers in this area.

**Review Assessment: Checking Correctness Of Derivations And Theory:**

I assessed the sensibility of the derivations and theory.

**Review Assessment: Checking Correctness Of Experiments:**

I assessed the sensibility of the experiments.

**Review Assessment: Thoroughness In Paper Reading:**

I read the paper at least twice and used my best judgement in assessing the paper.

---

> ### Author Response · Authors · 2019-11-12
> **Response to Reviewer #2**
>
> We thank the reviewer for the thoughtful review and the points about clarity, novelty, and strong experimental quality. We address your concerns below, and clarify points raised in the review:
>
> > “Using previous generated tokens as the unlikely tokens for the current generation step in the token-level unlikelihood training is intuitive, but also seems too simple.”
>
> The simplicity can be viewed as a strength. It is desirable to have candidates that do not require additional supervision (previous context tokens are already present in the training sequence), are fast to compute, and are interpretable.
>
> > “How is the training looking like? Do we need Gumbel-softmax-like trick to backpropagate through the generated tokens in the sequence-level training? Or, this is not needed? Can the authors clarity the training process?”
>
> It is not complicated. The token-level training is similar to standard MLE training, just with the additional candidate and loss computation (which can be done efficiently with matrix operations, for full details the code is provided in the link above, and you can look in the candidate_penalty_ce_loss.py file).
>
> The sequence-level training consists of decoding a sequence from the model, then computing the unlikelihood loss using the decoded sequence instead of ground truth; again, it is simple, and does not require gumbel-softmax nor policy gradient, which is a benefit of our approach. For full details the implementation is provided in the link above in the sequence_penalty_loss.py file.
>
> > “The proposed model can be directly applied to dialog response generation task, which also requires diversity in generated responses.”
>
> Indeed, unlikelihood training is a general framework that can be applied to other sequence generation tasks, and this is an exciting area for future work.

---

### Official Review · AnonReviewer1 · 2019-10-27
**Official Blind Review #1**

**Rating:** 3

**Review:**

This paper targets on solving the dull and repetitive outputs in MLE training for neural text generation. The authors propose a new unlikelyhood training to avoid assigning much probability to frequent and repetitive words. The authors combine the proposed algorithms and beam search and state that the results improved over beam blocking and neculus decoding.

The unlikelyhood training is to provide a set of negative candidates and minimize the probability of these tokens. This raises several practical issues: how to choose a reasonable $\alpha$.
This set can be chosen as the previous tokens in the sequence. This is a reasonable choice, but the author does not state why the other choices are not working,e.g. Sharpening the distribution using temperature. A potential counter case is that there are similar words exists in the sequences, but the unlikely loss trends to distinguish these synonyms.  The other unlikelyhood training choice is called sequence-level set. However, it seems not sequence-level but just n-gram center.  A question would be why not chose the whole n-gram instead of just choosing the center of n-gram. Also, why a prefix is really needed is questionable.


Eq 8 seems wrong, why$i \leq n \leq j$

Table 2 should have shown the original sequences on the repetition metrics to show it indeed make sense.ppl should be enough, acc seems redundant. It seems that unlikely training may be harmful to ppl, which is the common metric to evaluate generation quality. A better discussion should be made on this to explain why it performance or if ppl has some problem.

Table 3 comparison may not be reasonable. As Nucleus sampling and beam blocking is not in training phase. This comparison is not really fair.


**Experience Assessment:**

I have published one or two papers in this area.

**Review Assessment: Checking Correctness Of Derivations And Theory:**

I assessed the sensibility of the derivations and theory.

**Review Assessment: Checking Correctness Of Experiments:**

I assessed the sensibility of the experiments.

**Review Assessment: Thoroughness In Paper Reading:**

I read the paper at least twice and used my best judgement in assessing the paper.

---

> ### Author Response · Authors · 2019-11-12
> **Response to Reviewer #1**
>
> Reviewer #1 raised minor concerns and had a few misunderstandings which we clarify below; we emphasize that our proposal, unlikelihood training, and its strong empirical results are significant and we encourage the reviewer to re-evaluate the rating after considering these clarifications.
>
> > “This raises several practical issues: how to choose a reasonable \alpha.”
>
> We found that \alpha = 1.0 worked, and we did not have to perform a hyper-parameter search for \alpha, so it was not a practical issue. For further insight, the gradient analysis (eqn. 6) shows how \alpha can affect the gradient, and we provide an interpretation with \alpha = 1.0.
>
> > “This is a reasonable choice, but the author does not state why the other choices are not working,e.g. Sharpening the distribution using temperature.”
>
> Unlikelihood is indeed a general framework, so there are different possibilities for choosing candidates. We show that previous context candidates result in substantial improvements, and provide an interpretation for why they work (see below eqn. 5). Can you clarify what you mean by “sharpening the distribution using temperature”, with respect to choosing candidates?
>
> > “A potential counter case is that there are similar words exists in the sequences, but the unlikely loss trends to distinguish these synonyms.”
>
> We have not observed evidence for this counter case as seen in the human evaluation of Table 3 or examples in Table 1; if the model was resorting to a simplistic strategy, it is unlikely that the human evaluation quality would show such substantial gains over the MLE baseline.
>
> > “The other unlikelyhood training choice is called sequence-level set. However, it seems not sequence-level but just n-gram center.”
>
> This is a misunderstanding we should clarify. Sequence-level refers to the fact that the loss is computed over a model-decoded sequence. A token is penalized if it is in any part of a repeating n-gram, not just the center of the n-gram (this is the intention of equation 8).
>
> We will clarify equation (8) by changing it to “x_t is the (single) negative candidate for step t if it is part of an n-gram that previously occurred in x_{<t}”. These are minor clarifications, and our method and results do not change.
>
> > “Also, why a prefix is really needed is questionable.”
>
> The task is sequence completion, which uses a prefix (defined in section 3).
>
> > “Table 2 should have shown the original sequences on the repetition metrics to show it indeed make sense.”
>
> The completions are not necessarily supposed to match the human completion, just resemble a human completion. A full evaluation of this question is done in the human evaluation (Table 3).
>
> > “Ppl would be enough, acc seems redundant.”
>
> Next-token prediction accuracy (acc) more closely measures the model’s ability to highly rank the top token instead of modeling the full distribution. We report both to provide as much information as possible in interpreting our results.
>
> > Re: “It seems that unlikely training may be harmful to ppl, which is the common metric to evaluate generation quality. A better discussion should be made on this to explain why it performance or if ppl has some problem.”
>
> In Table 2, the perplexity did not degrade after sequence-level training, actually it’s slightly better than the MLE baseline (L_{UL-seq} 25.42 vs. L_{MLE} 25.64), while having substantially better generations (e.g. seq-rep-4, 71% human eval win rate).
>
> We comment on this phenomenon in the paper: “the likelihood objective is not constrained enough, in the sense that two models with the same perplexity can exhibit wildly difference generation performance”, and perform human evaluations to quantify the phenomenon.
>
> >“Table 3 comparison may not be reasonable. As Nucleus sampling and beam blocking is not in training phase. This comparison is not really fair.”
>
> Nucleus sampling and beam blocking are not training-time methods (https://arxiv.org/abs/1904.09751, https://arxiv.org/abs/1705.04304). They are used only at inference time given a trained model. This is in fact a motivation of this paper: investigating a training-time method to address text degeneration.

---

### Decision · Program_Chairs · 2019-12-19

**Decision:**

Accept (Poster)

**Comment:**

This paper introduces a new objective for text generation with neural nets.  The main insight is that the standard likelihood objective assigns excessive probability to sequences containing repeated and frequent words.  The paper proposes an objective that penalizes these patterns.  This technique yields better text generation than alternative methods according to human evaluations.

The reviewers found the paper to be written clearly. They found the problem to be relevant and found the proposed solution method to be both novel and simple.  The experiments were carefully designed and the results were convincing.  The reviewers raised several concerns on particular details of the method.  These concerns were largely addressed by the authors in their response.  Overall, the reviewers did not find the weaknesses of the paper to be serious flaws.

This paper should be published. The paper provides a clearly presented solution for a relevant problem, along with careful experiments.